# A Disintegrin and Metalloprotease 12 Promotes Tumor Progression by Inhibiting Apoptosis in Human Colorectal Cancer

**DOI:** 10.3390/cancers13081927

**Published:** 2021-04-16

**Authors:** Young-Lan Park, Sun-Young Park, Hyung-Hoon Oh, Min-Woo Chung, Ji-Yun Hong, Ki-Hyun Kim, Dae-Seong Myung, Sung-Bum Cho, Wan-Sik Lee, Hyun-Soo Kim, Young-Eun Joo

**Affiliations:** Department of Internal Medicine, Chonnam National University Medical School, Gwangju 61469, Korea; kokodeak@hanmail.net (Y.-L.P.); sunfrog@hanmail.net (S.-Y.P.); hyung1125@naver.com (H.-H.O.); zeinf@hanmail.net (M.-W.C.); hongjymd@gmail.com (J.-Y.H.); forestinmind@hanmail.net (K.-H.K.); myungdaeseong@hanmail.net (D.-S.M.); eunabumi@chonnam.ac.kr (S.-B.C.); iliad@chonnam.ac.kr (W.-S.L.); dshskim@chonnam.ac.kr (H.-S.K.)

**Keywords:** ADAM12, cell survival, prognosis, colorectal neoplasm

## Abstract

**Simple Summary:**

A disintegrin and metalloprotease 12 (ADAM12) has been associated with tumor development and progression. The aim of the current study was to evaluate the impact of ADAM12 on cancer progression, prognosis, and therapeutic targets in colorectal cancer (CRC). Our results show that ADAM12 overexpression enhanced proliferation, inhibited apoptosis, and acted as a positive regulator of cell cycle progression in CRC cells. Phosphorylation of phosphatase and tensin homolog deleted on chromosome 10 (PTEN) was decreased and that of Akt was increased by ADAM12 overexpression. These results were reversed upon ADAM12 knockdown. ADAM12 overexpression was significantly associated with the cancer stage, depth of invasion, lymph node metastasis, distant metastasis, and poor survival in CRC patients. In a mouse xenograft model, tumor area, volume, and weight were significantly greater for the ADAM12 overexpression group and significantly lower for the ADAM12 knockdown group. In conclusion, ADAM12 may serve as a promising biomarker and/or therapeutic target in CRC.

**Abstract:**

A disintegrin and metalloprotease 12 (ADAM12) has been implicated in cell growth, tumor formation, and metastasis. Therefore, we evaluated the role of ADAM12 in colorectal cancer (CRC) progression and prognosis, and elucidated whether targeted downregulation of ADAM12 could lead to therapeutic sensitization. The effect of ADAM12 on tumor cell behavior was assessed in CRC cell lines, CRC tissues, and a mouse xenograft model. ADAM12 overexpression enhanced proliferation, inhibited apoptosis, and acted as positive regulator of cell cycle progression in CRC cells. Phosphorylation of PTEN was decreased and that of Akt was increased by ADAM12 overexpression. These results were reversed upon ADAM12 knockdown. ADAM12 overexpression was significantly associated with the cancer stage, depth of invasion, lymph node metastasis, distant metastasis, and poor survival in CRC patients. In a mouse xenograft model, tumor area, volume, and weight were significantly greater for the ADAM12-pcDNA6-myc-transfected group than for the empty-pcDNA6-myc-transfected group, and significantly lower for the ADAM12-pGFP-C-shLenti-transfected group than for the scrambled pGFP-C-shLenti-transfected group. In conclusion, ADAM12 overexpression is essential for the growth and progression of CRC. Furthermore, ADAM12 knockdown reveals potent anti-tumor activity in a mouse xenograft model. Thus, ADAM12 may serve as a promising biomarker and/or therapeutic target in CRC.

## 1. Introduction

Colorectal cancer (CRC) is one of the major causes of cancer-associated morbidity and mortality worldwide. Despite significant advancements in diagnosis and therapy, fewer than 40% of cases are diagnosed when the cancer is still localized, and the prognosis of advanced CRC remains poor, with progressive behaviors, including cancer invasion and metastasis, being the major contributors to CRC-related morbidity and mortality [1,2,3]. This highlights the need to develop biomarkers to identify which persons will benefit the most from cancer surveillance and management. Identifying biomarkers that can detect CRC early or monitor cancer progression enables us to personalize medicine and improve the survival rates of patients with cancer [4,5]. 

A disintegrin and metalloproteases (ADAMs) are members of the membrane-anchored metzincin superfamily of zinc-based metalloproteinases. The mammalian ADAM protein family consists of 40 members, according to current research, and contains a metalloproteinase and disintegrin domain that combines proteolytic and adhesive functions [6,7]. This family is involved in various physiological and pathological processes, including cellular development, modulation of inflammatory reactions, and formation and progression of cancer via the release of membrane-bound proteins, such as adhesion molecules, cytokines, chemokines, and growth factors [8,9,10]. 

The human ADAM12 gene is located at 10q26.3 and has a typical ADAM family structure sequence, including metalloproteinase, disintegrin, cysteine-rich, and epidermal growth factor-like domains. ADAM12 is predominantly expressed in certain tissues, including bone, cartilage, brain, liver, heart, and muscle. ADAM12 is a proteolytic member of the ADAM protein family whose functions have been implicated in fertilization, cell adhesion, fusion, migration, proteolysis of extracellular matrix, and cell signaling [11,12]. Recently, ADAM12 has been reported to be highly expressed in various human cancers, including breast, bladder, and lung cancers, and has been associated with tumor development and progression [13,14,15,16,17,18,19,20,21,22,23,24]. ADAM12 has also been shown to regulate tumor progression in mouse tumor models, either by enhancing resistance to tumor cell apoptosis, by increasing tumor cell proliferation and invasion, or by promoting epithelial-to-mesenchymal transition [25]. In addition, ADAM12 regulates a variety of cellular processes in cancers via multiple signaling transduction pathways, including Akt [11,12]. Previous studies have shown upregulation of ADAM12 in CRC, but its clinical and functional relevance has not been explored [26,27,28]. 

The aims of the current study were to evaluate the role of ADAM12 in human CRC progression and prognosis and to elucidate whether targeted downregulation of ADAM12 could lead to therapeutic sensitization using an in vivo mouse xenograft model. 

## 2. Results

### 2.1. Expression of ADAM12 in Human CRC Cells 

To investigate the expression of ADAM12 in CRC cells, ADAM12 protein expression was examined using Western blotting with ADAM12 and Myc-tagging antibodies in human CRC cell lines DLD1 and SW480. ADAM12 expression was not observed in DLD1 and SW480 cells. Therefore, we experimented with transfecting the ADAM12 gene and knocking it out in DLD1 and SW480 cells. ADAM12-pcDNA6-myc construct or ADAM12 siRNA was used to transfect the ADAM12 gene and knock it out in DLD1 and SW480 cells. ADAM12 protein expression in tested cells showed a significant increase at the protein level by the transfection of ADAM12-pcDNA6-myc construct and a significant decrease by the transfection of ADAM12 siRNA (Figure 1A and Appendix A). 

### 2.2. Effect of ADAM12 on Proliferation of Human CRC Cells

To determine the potential effects of ADAM12 on cell proliferation, cells were subjected to a cell proliferation assay two days after transfection with ADAM12-pcDNA6-myc construct or ADAM12 siRNA. The number of proliferating cells, determined by measuring the absorbance of cell suspension, significantly increased to a greater extent in the ADAM12-pcDNA6-myc-transfected DLD1 and SW480 cells than in the empty-pcDNA6-myc-transfected cells (*p* < 0.05 and *p* < 0.05, respectively). In contrast, ADAM12 siRNA-transfected DLD1 and SW480 cells showed significantly lower proliferation compared to scrambled siRNA-transfected cells (*p* < 0.05 and *p* < 0.05, respectively; Figure 1B). Plate colony formation assay showed significantly increased cell proliferation due to ADAM12 overexpression (*p* < 0.05 and *p* < 0.05, respectively) and decreased cell proliferation due to ADAM12 knockdown (*p* < 0.05 and *p* < 0.05, respectively; Figure 1C). Additionally, the bromodeoxyuridine (BrdU) incorporation of proliferating cells was significantly increased in the ADAM12-pcDNA6-myc-transfected DLD1 and SW480 cells compared to the empty-pcDNA6-myc-transfected cells (*p* < 0.05 and *p* < 0.05, respectively). In contrast, ADAM12 siRNA-transfected DLD1 and SW480 cells showed significantly reduced proliferation compared to scrambled siRNA-transfected cells (*p* < 0.05 and *p* < 0.05, respectively; Figure 1D).

### 2.3. Effect of ADAM12 on Apoptosis and Cell Cycle Progression in Human CRC Cells

To evaluate the effect of ADAM12 on apoptosis and cell cycle progression, we performed flow cytometric analysis. The cell apoptotic rate induced by the transfection of ADAM12-pcDNA6-myc construct and ADAM12 siRNA significantly increased to a greater extent compared with the transfection of ADAM12-pcDNA6-myc construct and scrambled siRNA (*p* < 0.05 and *p* < 0.05, respectively) in DLD1 and SW480 cells (Figure 2A,B). To determine the activation of caspases during overexpression and knockdown of ADAM12, we further investigated caspase-specific activities. The expression of cleaved caspase-3, caspase-7, and poly (ADP-ribose) polymerase (PARP) was downregulated after overexpression and upregulated after knockdown of ADAM12 in DLD1 and SW480 cells (Figure 2C and Appendix A). 

Overexpression of ADAM12 decreased the apoptotic fraction (sub-G1 phase) and knockdown of ADAM12 promoted the apoptotic fraction of DLD1 and SW480 cells (Figure 3A). Next, we evaluated the effects of ADAM12 on cyclins, cyclin-dependent kinases (CDKs), and CDK inhibitors involved in cell cycle progression. As shown in Figure 3B, cyclin D1 and CDK6 protein levels were increased by ADAM12 overexpression in DLD1 and SW480 cells. In contrast, cyclin D1 and CDK6 protein levels were decreased in response to ADAM12 knockdown. p21 and p27 protein levels were decreased by ADAM12 overexpression in DLD1 and SW480 cells. In contrast, p21 and p27 protein levels were increased in response to ADAM12 knockdown (Figure 3B and Appendix A).

### 2.4. Effect of ADAM12 on Oncogenic Signaling Pathways in Human CRC Cells

To examine whether ADAM12 activates the intracellular signaling pathways in human CRC cells, we determined the phosphorylation levels of Akt and phosphatase and tensin homolog deleted on chromosome 10 (PTEN) signaling proteins using Western blotting. We found that the phosphorylation of PTEN was decreased and the Akt level was increased due to the overexpression of ADAM12 in DLD1 and SW480 cells. In contrast, the phosphorylation of PTEN was increased and that of Akt was decreased by the knockdown of ADAM12 (Figure 4 and Appendix A).

### 2.5. Expression of ADAM12 in Human CRC Tissues 

We evaluated ADAM12 expression at the mRNA and protein level using RT-PCR and immunohistochemistry in human CRC tissues and in normal colorectal mucosal tissue counterparts from the same patients, collected using colonoscopic biopsy and surgical specimens. In the colonoscopic biopsy specimens, we confirmed the upregulation of ADAM12 expression to higher levels in human CRC tissues than in normal mucosa counterparts at the mRNA level (Figure 5A). In paraffin tissue sections, immunostaining of ADAM12 protein was predominantly identified in the cytoplasm of cancer cells and was not detectable in the tumor stroma. Immunostaining showed that ADAM12 protein was not stained or weakly stained in the normal colorectal mucosa (Figure 5B). 

### 2.6. Association of ADAM12 with Clinicopathological Variables of Human CRC

To study the prognostic role of ADAM12 in human CRC progression, we immunohistochemically investigated the expression of ADAM12 protein in formalin-fixed, paraffin-embedded tissue blocks obtained from 366 CRC patients. The correlation between ADAM12 immunostaining results and the clinicopathological parameters of patients was analyzed. We observed that ADAM12 expression was significantly associated with cancer stage, depth of invasion, lymph node metastasis, and distant metastasis (*p* < 0.001, *p* = 0.026, *p* = 0.003, and *p* < 0.001, respectively; Table 1). Moreover, the overall survival of patients with ADAM12-positive tumors was significantly lower than that of patients with ADAM12-negative tumors (*p* = 0.001) (Figure 6). 

### 2.7. Correlation between ADAM12 Protein Expression and Tumor Cell Survival in Human CRC

All tumor samples underwent terminal deoxynucleotidyl transferase-mediated dUTP nick-end labeling (TUNEL) assay and immunostaining for Ki-67 to identify tumor cell survival. The apoptotic index (AI) of the 366 tumors ranged from 0.6 to 30.0, with a mean of 8.8 ± 5.6. The mean AI value of ADAM12-positive tumors was 6.4 ± 3.7, which was significantly lower than that of ADAM12-negative tumors (*p* = 0.009). The Ki-67 labeling index (KI) of the 366 tumors ranged from 21.9 to 59.7 with a mean of 52.8 ± 15.1. There was no significant difference between ADAM expression and KI results (*p* = 0.504; Table 2).

### 2.8. Effect of ADAM12 on Tumorigenesis of Human CRC Cells in In Vivo Mouse Xenograft Model

Given our in vitro and human clinical outcome data, we tested the role of ADAM12 in tumor initiation and proliferation in human CRC SW480 cells in vivo, by injecting the same number of empty-pcDNA6-myc vector, ADAM12-pcDNA6-myc construct, scrambled pGFP-C-shLenti, or ADAM12-pGFP-C-shLenti-transfected SW480 cells into non-obese diabetic/severe combined immunodeficiency (NOD/SCID) mice. After 37 days, the tumor area, volume, and weight of the mice in the ADAM12-pcDNA6-myc-transfected group were significantly greater than those of the mice in the empty-pcDNA6-myc-transfected group (Figure 7A,B). The tumor area, volume, and weight for the ADAM12-pGFP-C-shLenti-transfected group were significantly lower compared to the scrambled pGFP-C-shLenti-transfected group (Figure 7A,B). Taken together, these results indicate that ADAM12 is essential for the growth and progression of human CRC cells in vivo, as well as in vitro.

## 3. Discussion

Altered expression of ADAM12 is implicated in a variety of pathological diseases, such as diabetes, sepsis, rheumatoid arthritis, Alzheimer’s disease, atherosclerosis, cardiovascular disease, asthma, and cancer [11,12]. ADAM12 expression is increased in many human cancers, such as breast, bladder, lung, prostate, and liver cancer, and its expression correlates with the tumor stage and prognosis in breast, bladder, and lung cancer [13,14,15,16,17,18,19,20,21,22]. However, its role and molecular mechanisms in CRC remain unclear [26,27,28]. Therefore, we investigated whether ADAM12 may serve as a novel biomarker and/or therapeutic target in CRC, using human CRC cell lines, human CRC tissues, and an in vivo xenograft tumor model.

Proper regulation of cell proliferation, apoptosis, and cell cycle is crucial in order to maintain tissue homeostasis, and its dysregulation is a principal hallmark of cancer cells [29,30]. Furthermore, cancer cells are typically characterized by increased proliferation and resistance toward apoptosis and cell cycle control [31,32]. Previously, upregulation of ADAM12 was shown to promote proliferation and inhibit apoptosis of various human cancer cells [13,14,15,16,17,18,19,20,21,22]. In our study, overexpression of ADAM12 enhanced proliferation and inhibited apoptosis by downregulating caspase-specific activities, and acted as a positive regulator of cell cycle progression via the upregulation of cyclins and CDKs and the downregulation of CDK inhibitors in human CRC cells. These results were reversed after knockdown of ADAM12. These results suggest that ADAM12 contributes to the alteration of tumor cell survival in human CRC.

To elucidate the underlying mechanisms leading to these results, we analyzed the effect of ADAM12 on the stimulation of multiple intracellular signaling pathways regulating tumor cell survival. The Akt signaling pathway is involved in many cellular programs, such as cell survival and proliferation. Dysregulation of this pathway has been associated with many human diseases, including cancer, highlighting its potential as a therapeutic target for cancer [33]. The tumor suppressor PTEN is involved in the regulation of a variety of pathophysiological processes, such as cell proliferation, adhesion, and apoptosis, through inactivation of focal adhesion kinase and downregulation of the Akt signaling pathway [34]. Previously, it was shown that ADAM12 inhibits apoptosis by regulating Akt and mitogen-activated protein kinase signaling pathways in breast and liver cancer cells [22,35,36]. Additionally, ADAM12 regulates prostatic cancer cell invasion through the NF-κB signaling pathway [37]. Consistent with these findings, we found that phosphorylation of PTEN was decreased and Akt was increased by overexpression of ADAM12 in human CRC cells. In contrast, these results were reversed after knockdown of ADAM12. PTEN, a dual protein and lipid phosphatase, dephosphorylates not only proteins, but also phosphatidylinositol-3,4,5-trisphosphate (PIP3), generated by PI3-kinase, thus counteracting the Akt signaling pathway. Thus, loss of PTEN could lead to increased amounts of PIP3, which is a strong activator of the Akt signaling pathway [33,34]. These results suggest that ADAM12 participates in altering the oncogenic behavior of human CRC cells by regulating the phosphorylation of Akt and PTEN. 

In our study, although ADAM12 expression was not observed in CRC cell lines, ADAM12 expression was upregulated to higher levels in human CRC tissues than in normal mucosa counterparts at the mRNA and protein levels in colonoscopic biopsy and surgical specimens. Overexpression of ADAM12 was significantly associated with the cancer stage, depth of invasion, lymph node metastasis, distant metastasis, and poor survival in patients with CRC. Moreover, we observed that the mean AI value of ADAM12-positive tumors was significantly lower than that of ADAM12-negative tumors. These results confirm the resisting apoptotic potential of ADAM12 in vivo, consistent with the results of in vitro studies. 

Finally, given our in vitro and human clinical outcome data, we evaluated the effect of ADAM12 on the tumorigenesis of human CRC cells in an in vivo xenograft model with human CRC cells subcutaneously injected into immunocompromised mice. The tumor area, volume, and weight in the ADAM12-pcDNA6-myc-transfected mice were significantly greater than those of the empty-pcDNA6-myc-transfected mice. The tumor area, volume, and weight in the ADAM12-pGFP-C-shLenti-transfected mice were significantly reduced compared to those in the scrambled pGFP-C-shLenti-transfected mice. These results indicate that the ADAM12 status is associated with cancer growth and progression in the in vivo mouse xenograft model.

However, the subcutaneous xenograft model used in our study has limitations: the lack of a direct relation with local invasion and metastases, and the ability to reproduce the tumor/microenvironment interaction. Therefore, orthotopic implantation of human CRC cells into the equivalent colons of live mice is preferred to evaluate the in vivo efficacy of new biomarkers and therapeutic targets in CRC because it provides tumor cells a microenvironment for organotypic interaction [38]. Further study using an orthotopic xenograft model is needed to validate ADAM12 as a biomarker and therapeutic target for CRC.

## 4. Materials and Methods 

### 4.1. Cell Culture

Human colorectal cancer (CRC) cell lines (SW480, CCL-228TM and DLD1, CCL-221TM) obtained from patients with CRC adenocarcinoma at Dukes stage B and C were purchased from the American Type Culture Collection (Manassas, VA, USA). Cells were cultured in high glucose Dulbecco’s modified Eagle’s medium (DMEM; Gibco, Thermo Fisher Scientific, Inc., Waltham, MA, USA) supplemented with 10% fetal bovine serum (Gibco, Thermo Fisher Scientific, Inc.). All cells were maintained at 37 °C in a humidified atmosphere containing 5% CO_2_.

### 4.2. Gene Transfection

The ADAM12-pcDNA6-myc vector was constructed to overexpress the ADAM12 gene in both in vitro and in vivo experiments. To silence the gene, ADAM12 small interfering (si) RNA (GACUACAACGGGAAAGCAA-dTdT) and ADAM12-pGFP-C-shLenti construct were purchased from Bioneer (Daejeon, Korea) and Origene (Rockville, MD, USA), respectively. Transfection of the vector and siRNAs was performed using Lipofectamine^TM^ 2000 and Lipofectamine^TM^ RNAiMAX (Invitrogen, Thermo Fisher Scientific, Inc.), according to the manufacturer’s instructions. Stable transfectants containing the ADAM12-pcDNA6-myc construct and the ADAM12-pGFP-C-shLenti construct were isolated using DMEM with antibiotics (10 μg/mL blasticidine and 2 μg/mL puromycin, respectively).

### 4.3. Cell Proliferation Assay

Gene-transfected cells were plated onto 96-well culture plates at a density of 5000 cells per well in triplicate. At the indicated time points, 10% of water-soluble tetrazolium salt reagent (WST-1; EZ-CYTOX, DoGen; Daeillab, Seoul, South Korea) was added. After incubation for an additional 2 h, the optical density (OD) was measured at 450 nm using an Infinite M200 (Tecan, Mannedorf, Switzerland). All experiments were performed 3 times.

### 4.4. Plate Colony Formation Assay 

Cells were seeded in a 6-well plate at a density of 400 cells/well after 24 h of gene transfection and maintained for 8 days at 37 °C in humidified 5% CO_2_ conditions. Cells were washed with phosphate-buffered saline (PBS) to remove the debris and fixed using 4% paraformaldehyde for 1 h. The plates were then washed with PBS and stained with a 0.1% crystal violet solution (Sigma-Aldrich, St. Louis, MO, USA) for 5 min. The number of colonies with diameters larger than 1 mm was counted after washing.

### 4.5. Bromodeoxyuridine (BrdU) Cell Proliferation Assay

BrdU Cell Proliferation Assay Kit (cell signaling) was used for the BrdU incorporation assay. In brief, cells labeled with BrdU for 24 h were treated with fixing/denaturing solution, then with mouse anti-BrdU for 1 h. After incubation with horseradish peroxidase (HRP)-conjugated antibody, quantified BrdU-labeled cells were added to the tetramethylbenzidine (TMB) substrate and measured using a microplate reader.

### 4.6. Reverse-Transcription PCR

Total RNA was extracted from the biopsy tissues using Trizol reagent (Invitrogen, Carlsbad, CA, USA) and cDNA was reverse transcribed from 500 ng of total RNA using PrimeScript^®^ RT Master Mix (Takara, Japan). The PCR reactions were carried out using specific primers. The following primers were used: human ADAM12 (forward-GCTGATGAAGTTGTCAGTGC, reverse-CATGACAATTCCCCCAGACTG) and housekeeping gene glyceraldehyde 3-phosphate dehydrogenase (GAPDH) (forward-ACCACAGTCCATGCCATCAC, reverse-TCCACCACCCTGTTGCTGTA). The reaction products were visualized on 1% agarose electrophoresis gel using EtBr staining, and the bands were analyzed using Multi Gauge version 3.0 (Fujifilm, Tokyo, Japan).

### 4.7. Western Blotting

The gene-transfected cells were collected, washed with cold PBS, and lysed in radioimmunoprecipitation assay (RIPA) buffer (Thermo Fisher Scientific, Inc.) with Halt^TM^ Protease and Phosphatase Inhibitor Cocktail (Thermo Fisher Scientific, Inc.). The protein concentration was quantified using BCA^TM^ protein assay (Thermo Fisher Scientific, Inc.). The total cellular lysates were resolved using sodium dodecyl sulfate (SDS)–polyacrylamide gel electrophoresis, subsequently transferred onto a polyvinylidene fluoride (PVDF) membrane (Millipore, Billerica, MA, USA), and separately immunoblotted with specific antibodies at 1:1000 dilution. The following antibodies were used: ADAM12 (Abcam, Cambridge, UK), Myc-tag, cleaved poly (ADP-ribose) polymerase (PARP), cleaved caspase-3, cleaved caspase-7, cyclin D1, p27Kip1, cyclin-dependent kinase 6 (CDK6), p21 Waf1/Cip1, phospho-Akt (S473), Akt, phosphatase and tensin homolog deleted on chromosome 10 (PTEN) (S380), PTEN (Cell Signaling Technology, Inc., Danvers, MA, USA), and GAPDH (Santa Cruz Biotechnology, Inc., Dallas, TX, USA). The membrane was then washed and developed using an enhanced chemiluminescence detection kit (Amersham, GE Healthcare Life Sciences). The protein bands were visualized using an LAS-4000 luminescent image analyzer (Fujifilm, Tokyo, Japan). 

### 4.8. Flow Cytometry Determination of Apoptosis 

Cells were washed twice with PBS, then the pellets were resuspended in 1 × Binding Buffer (0.01 M HEPES pH 7.4, 0.14 M NaCl and 2.5 mM CaCl_2_). Each sample was transferred to a tube and stained with Annexin V-APC and 7-amino-actinomycin D (7-AAD) (BD Biosciences, San Diego, CA, USA) at room temperature for 20 min. After incubation in the dark at room temperature, 400 μL of 1× Binding Buffer was added and the cell suspensions were analyzed using a FACS Calibur flow cytometer (Becton Dickinson, San Jose, CA, USA) and WinMDI version 2.9 (Scripps Research Institute, Jupiter, FL, USA). Finally, 3 experiments were performed to analyze the rate of cell apoptosis.

### 4.9. Flow Cytometry Analysis of Cell Cycle Arrest

Cells were washed twice with PBS and fixed in ice-cold 70% ethanol. Fixed cells were incubated in 10 μg/mL ribonuclease A (Sigma-Aldrich, St. Louis, MO, USA) and stained with propidium iodide (50 mg/mL, Sigma-Aldrich) at room temperature for 30 min. Cells were subsequently analyzed using a FACS Calibur flow cytometer (Becton Dickinson). The fractions of cells in the sub-G1, G0/G1, S, and G2/M phases were analyzed using WinMDI version 2.9 (Scripps Research Institute Jupiter, FL, USA).

### 4.10. Patients and Tissue Samples

For the RNA preparation, 20 colon cancer tissue samples and normal colon tissue counterparts were collected via colonoscopic biopsy at Chonnam University Hwasun Hospital (Jeonnam, Korea). For immunohistochemistry (IHC), formalin-fixed and paraffin-embedded tissue samples were selected from 366 patients who underwent surgery for pathologically confirmed colon cancer at that hospital between January 2004 and December 2006. The primary selection criteria were the availability of formalin-fixed and paraffin-embedded tissue blocks and sufficient clinical follow-up for tumor-specific survival analysis. No patient had received preoperative chemo- or radiotherapy. Tissue blocks were selected by viewing the original pathologic slides and choosing blocks that showed the junction between the normal colon epithelium and the tumor region. Patients were followed up to December 2012, and overall survival (OS) was defined as the interval between surgery and death or the last follow-up. This study was approved by the Institutional Review Board of the Chonnam National University Hwasun Hospital (IRB No. CNUHH-2017-164), and written informed consent was obtained from every patient.

### 4.11. Immunohistochemistry (IHC)

Paraffin sections were deparaffinized and the endogenous peroxidase was neutralized using Peroxidase-Blocking Solution (Dako, Carpinteria, CA, USA). After antigen retrieval was performed in citrate buffer at pH 6.0, the tissues were incubated with the primary anti-ADAM12 (Abcam), Ki-67 (Dakopatts, Glostrup, Denmark). The tissue was then stained using the Dako REAL^TM^ EnVision HRP/DAB Detection System and counterstained with hematoxylin. 

### 4.12. Evaluation of ADAM12 Expression

The ADAM12 IHC score was calculated by multiplying the staining intensity score by the percentage of positive cells. The intensity of ADAM12 immunoreactive intensity was scored as follows: 0, negative; 1, weak; 2, moderate; and 3, strong. The scores for the percentage of positive cells were also divided into 4 grades: 1, 0–25%; 2, 26–50%; 3, 51–75%; and 4, >75% positive cells. The average of the final scores for the 366 tumor samples was presented as a reference point to divide them into 2 subgroups, positive and negative expression. Samples with a final score > 7 were designated as positive for ADAM12 expression. These judgments were made by 2 independent pathologists without knowledge of the patient clinical outcome data. 

### 4.13. Assessment of Apoptosis and Tumor Cell Proliferation 

Detection of apoptotic tumor cells in the sample tissue was performed using a terminal deoxynucleotidyl transferase-mediated dUTP nick-end labeling (TUNEL) technology system (Promega, Madison, WI, USA), according to the manufacturer’s instructions. Briefly, deparaffinized tissues were rehydrated and incubated in a permeabilization solution. Labeling of apoptotic cells was performed by adding terminal deoxynucleotide transferase enzyme (TdT) reaction mix to the tissue sections on the slides. After washing, color development for localization of labeled cells was performed by incubating the slides with the enzyme substrate 3,3-diaminobenzidine (DAB). The apoptotic index (AI) was determined by counting the number of TUNEL positive cells per 1000 tumor cell nuclei. Tumor proliferative cells were detected using Ki-67 antibody by IHC, and the immunostained nuclei were considered as positively labeled. The number of Ki-67 positive nuclei per 1000 tumor cell nuclei was presented as the Ki-67 labeling index (KI).

### 4.14. In Vivo Tumor Model Experiment

The mouse experimental design and protocols used in this study were approved by the Chonnam National University Hwasun Hospital animal care and use committee (CNUIACUC-H-2018-64). Male 6-week-old non-obese diabetic/severe combined immunodeficiency (NOD/SCID) mice were purchased from Charles River Laboratories (Yokohama, Japan) and divided into 4 groups (*n* = 6). Mice were allowed to adapt to the breeding environment for 7 days before the experiment, and were divided into 4 groups (each *n* = 6) for tumor cell implantation: empty-pcDNA6-myc vector transfected stable cells (EV, group 1), ADAM12-pcDNA 6-myc construct transfected stable cells (AV, group 2), AV and scrambled pGFP-C-shLenti construct double transfected stable cells (AV+Ssh, group 3), and AV and ADAM12-pGFP-C-shLenti construct double transfected stable cells (AV+Ash, group 4). To generate subcutaneous human colon cancer xenografts, 2 × 10^6^ stable cells with ADAM12-pcDNA 6-myc construct or ADAM12-pGFP-C-shLenti construct were suspended in 100 μL of serum-free DMEM, and cell suspension was subcutaneously injected into the back of each mouse. Thirty-seven days after tumor cell inoculation, all mice were sacrificed and the subcutaneous tumors were removed. The weight and volume of implanted tumors was calculated, and the tumor tissue was immediately fixed in a 10% formalin solution, then embedded in paraffin for hematoxylin and eosin (H&E) staining.

### 4.15. Statistical Analysis

The association between ADAM12 expression and clinicopathological parameters was analyzed using χ^2^ test and Fisher’s exact test with SPSS version 20.0 software (IBM Corporation, Armonk, NY, USA). Kaplan–Meier analysis and log-rank tests were used to perform survival analysis. The associations between experimental groups were determined using ANOVA/Tukey test. A value of *p* < 0.05 was considered significant.

## 5. Conclusions

In conclusion, overexpression of ADAM12 contributes to the oncogenic phenotypes of human CRC cells. Moreover, ADAM12 expression is upregulated in human CRC tissues and associated with poor prognosis in patients with CRC. In addition, overexpression of ADAM12 enhanced the growth and progression of human CRC cells, and knockdown of ADAM12 showed potent anti-tumor activity in our in vivo mouse xenograft model. Thus, ADAM12 may serve as a promising biomarker and/or therapeutic target in human CRC.

## Figures and Tables

**Figure 1 cancers-13-01927-f001:**
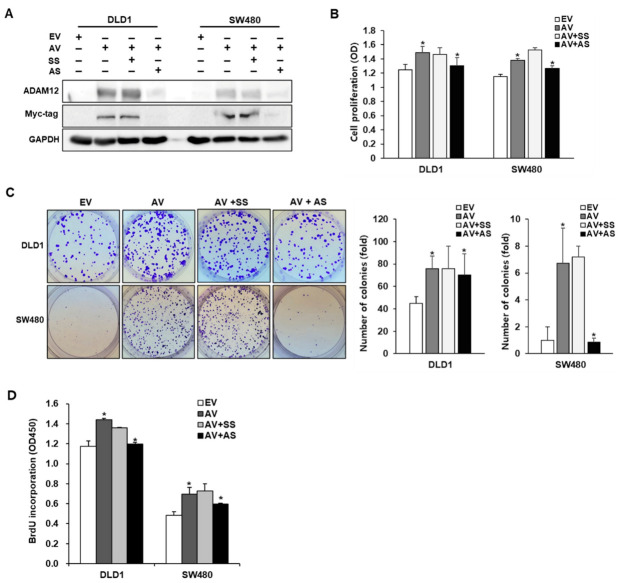
Effect of ADAM12 expression on cell proliferation of human CRC cells. (**A**) Expression of ADAM12 protein via gene transfection. Western blotting results revealed that ADAM12 protein was overexpressed and knocked down using ADAM12-pcDNA6-myc construct and ADAM12 siRNA, respectively. (**B**) Effect of ADAM12 expression on proliferation was determined using WST-1 assay in human CRC cells (** p* < 0.05). (**C**) Plate colony formation assay confirmed single-cell proliferative capability due to ADAM12 expression (** p* < 0.05). (**D**) Effect of ADAM12 expression on BrdU incorporation of proliferating human CRC cells (* *p* < 0.05). ADAM12, A disintegrin and metalloprotease 12; CRC, colorectal cancer; EV, EmptyV, empty-pcDNA6-myc vector; AV, ADAM12-pcDNA6-myc construct; SS, scramble siRNA; AS, ADAM12 siRNA; BrdU, bromodeoxyuridine.

**Figure 2 cancers-13-01927-f002:**
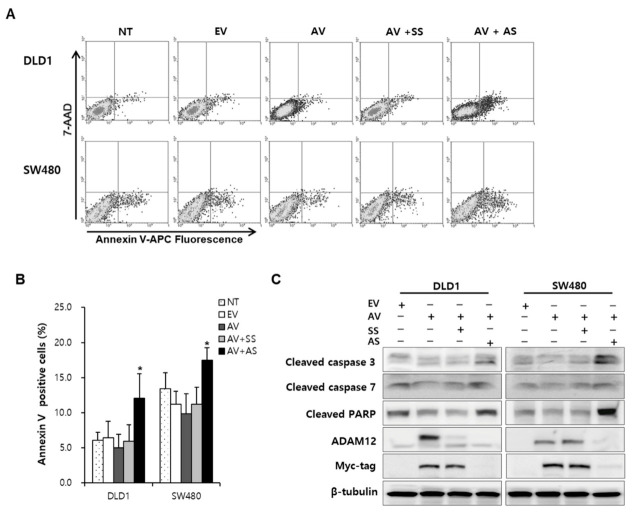
Effect of ADAM12 on apoptosis of human CRC cells. (**A**) Representative flow cytometry plots. (**B**) Cell apoptotic rate according to ADAM12 expression was analyzed using Annexin V-APC staining and a FACS Calibur flow cytometer. Bars represent mean ± standard deviation from three experiments (* *p* < 0.05). (**C**) Effect of apoptotic proteins according to ADAM12 expression was demonstrated using Western blotting and human CRC cells. Expression of cleaved caspase-3 and -7 and PARP was increased upon ADAM12 knockdown. ADAM12, A disintegrin and metalloprotease 12; CRC, colorectal cancer; NT, non-transfected cells; EV, EmptyV, empty-pcDNA6-myc vector; AV, ADAM12-pcDNA6-myc construct; SS, scramble siRNA; AS, ADAM12 siRNA; PARP, poly (ADP-ribose) polymerase.

**Figure 3 cancers-13-01927-f003:**
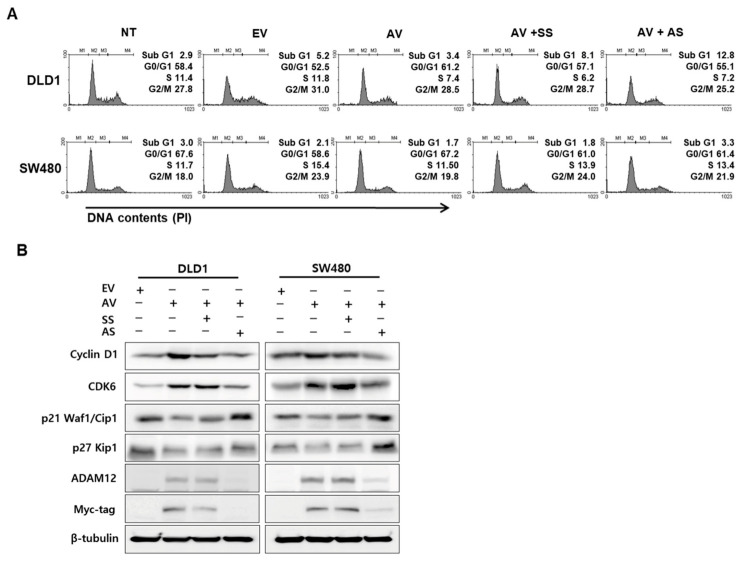
Effect of ADAM12 on cell cycle arrest in human CRC cells. (**A**) Transfected cells were stained with propidium iodide (PI), then cell cycle distribution by ADAM12 expression was analyzed using a FACS Calibur flow cytometer. (**B**) Effect of cell cycle-associated proteins based on ADAM12 expression was demonstrated using Western blotting. Cyclin D1 and CDK6 protein levels were increased by ADAM12 overexpression but decreased by ADAM12 knockdown. p21 Kip1 and p27 Waf1/Cip1 protein levels were reduced due to ADAM12 overexpression but increased by ADAM12 knockdown. ADAM12, A disintegrin and metalloprotease 12; CRC, colorectal cancer; NT, non-transfected cells; EV, EmptyV, empty-pcDNA6-myc vector; AV, ADAM12-pcDNA6-myc construct; SS, scramble siRNA; AS, ADAM12 siRNA; CDK6, cyclin-dependent kinase 6.

**Figure 4 cancers-13-01927-f004:**
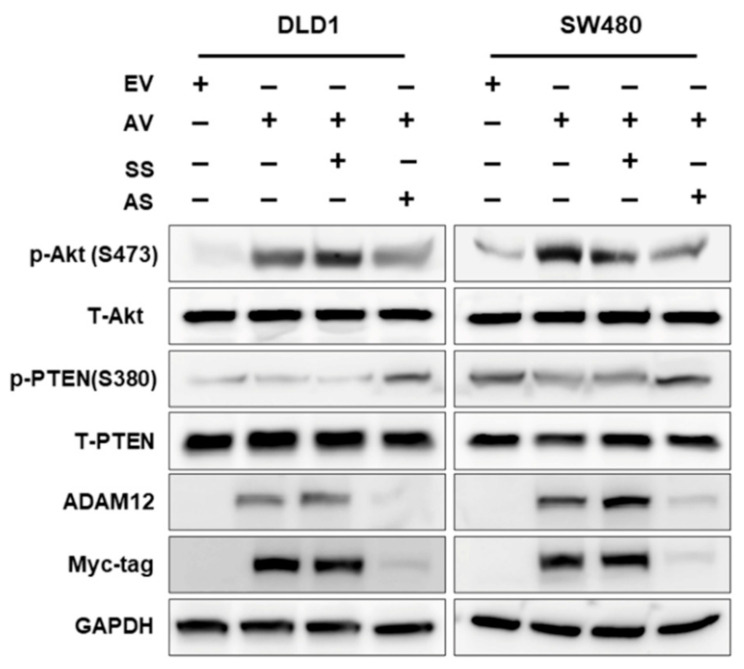
Effect of ADAM12 on oncogenic signaling pathways in human CRC cells. Western blotting demonstrated that phosphorylation of PTEN decreased and Akt level increased upon ADAM12 overexpression; opposite results were observed upon ADAM12 knockdown. ADAM12, A disintegrin and metalloprotease 12; CRC, colorectal cancer; PTEN, phosphatase and tensin homolog deleted on chromosome 10; EV, EmptyV, empty-pcDNA6-myc vector; AV, ADAM12-pcDNA6-myc construct; SS, scramble siRNA; AS, ADAM12 siRNA.

**Figure 5 cancers-13-01927-f005:**
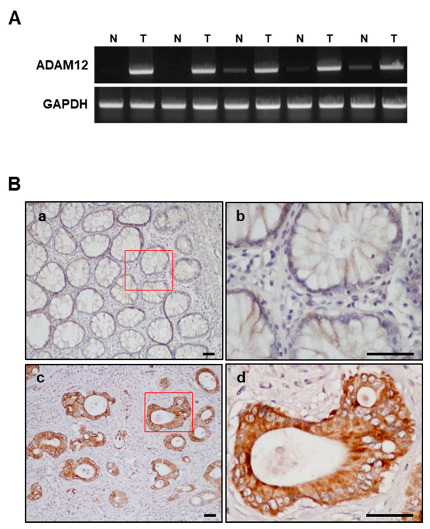
Expression of ADAM12 in human CRC tissues. (**A**) High expression of ADAM12 mRNA in CRC biopsy tissue was confirmed using RT-PCR. (**B**) High expression of ADAM12 protein in CRC tissue was confirmed using immunohistochemistry. a: normal colon mucosa; b: red square in a; c: CRC tissue; and d: red square in c. Scale bar = 10 μm. ADAM12, A disintegrin and metalloprotease 12; GAPDH, glyceraldehyde 3-phosphate dehydrogenase; CRC, colorectal cancer; N, normal colon mucosa; T, CRC tissue. ADAM12 immunoreactivity is predominantly identified in cytoplasm of tumor cells.

**Figure 6 cancers-13-01927-f006:**
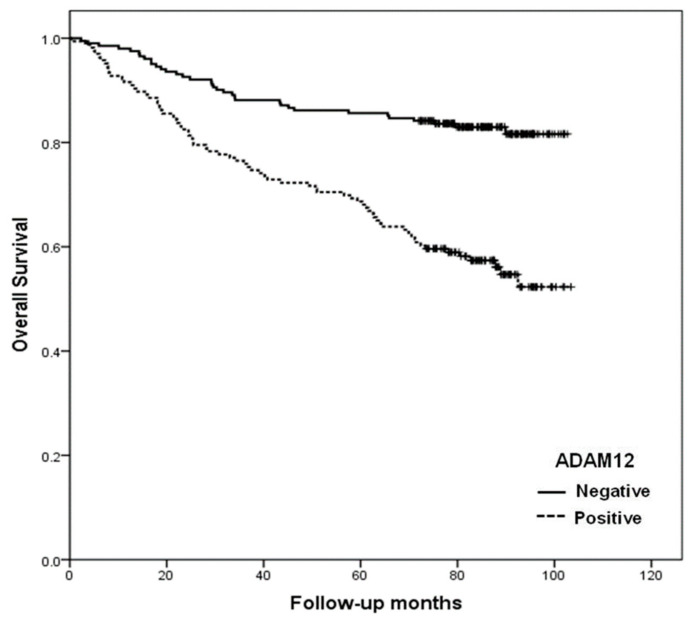
Overall survival curves according to ADAM12 expression in patients with CRC. Positive expression of ADAM12 (dotted line) was associated with poor survival in patients with CRC (*p* = 0.001). ADAM12, A disintegrin and metalloprotease 12; CRC, colorectal cancer.

**Figure 7 cancers-13-01927-f007:**
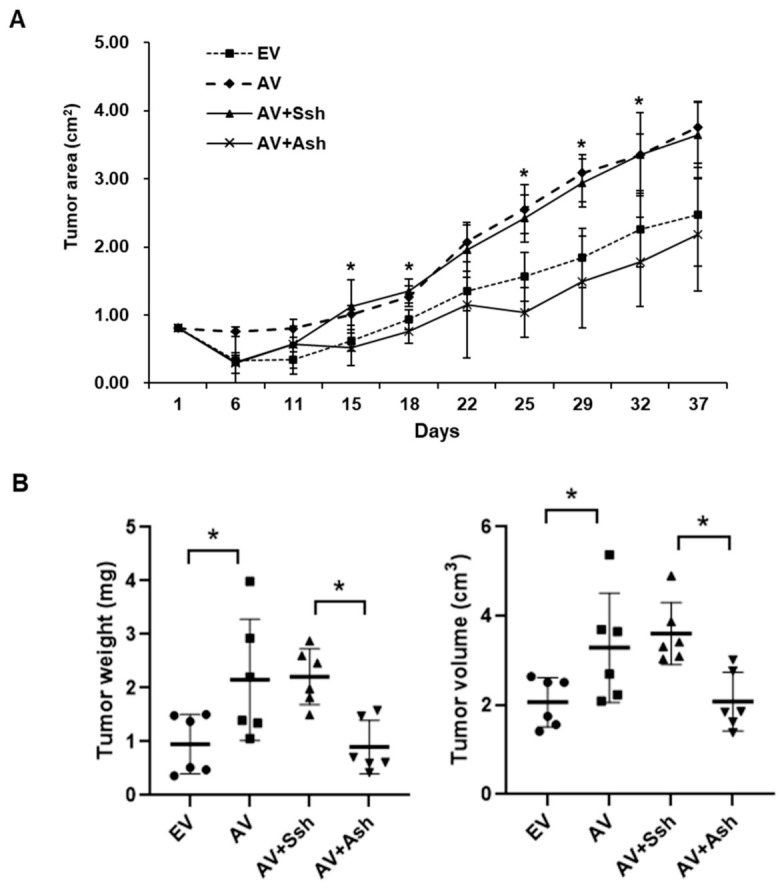
Effect of ADAM12 expression on tumor growth in subcutaneous xenograft model. SW480 cells with stable gene expression of ADAM12 were implanted subcutaneously onto backs of NOD/SCID mice (four groups, *n* = 6). (**A**) For 37 days, tumor growth was measured and calculated. (**B**) Volume and weight of tumor tissue obtained from sacrificed mice 37 days after transplantation were measured and calculated. ADAM12, A disintegrin and metalloprotease 12; NOD/SCID, non-obese diabetic/severe combined immunodeficiency; EV, EmptyV, empty-pcDNA6-myc vector; AV, ADAM12-pcDNA6-myc construct; Ssh, Scrambled pGFP-C-shLenti vector; Ash, ADAM12-pGFP-C-shLenti construct. * *p* < 0.05.

**Table 1 cancers-13-01927-t001:** Correlation between ADAM12 expression and clinicopathological parameters of human colorectal cancer.

Parameters		ADAM12 Expression		*p*-Value
	Total	Negative	Positive	
	(*n* = 366)	(*n* = 201)	(*n* = 165)	
Age (years)				0.229
<61.4	159	93	66	
≥61.4	207	108	99	
Sex				0.079
Male	220	129	91	
Female	146	72	74	
Tumor size (cm)				0.733
<4.9	201	112	89	
≥4.9	165	89	76	
Stage				<0.001
I	45	12	8	
II	143	89	54	
III	163	78	85	
IV	15	5	10	
Lymphovascular invasion				0.062
Negative	266	154	112	
Positive	100	47	53	
Perineural invasion				0.824
Negative	244	135	109	
Positive	122	66	56	
Histologic type				0.092
WD	127	66	61	
MD	191	103	88	
PD	30	17	13	
Mucinous/Signet ring cell	18	15	3	
Depth of invasion (T)				0.026
T1	15	12	3	
T2	47	30	17	
T3	289	154	135	
T4	15	5	10	
Lymph node metastasis (N)				0.003
N0	191	119	72	
N1-3	175	82	93	
Distant metastasis (M)				<0.001
M0	325	189	136	
M1	41	12	29	

ADAM12, A disintegrin and metalloprotease 12; WD, well differentiated; MD, moderately differentiated; PD, poorly differentiated.

**Table 2 cancers-13-01927-t002:** Correlation between ADAM12 expression and tumor cell proliferation or apoptosis in human colorectal cancer.

Indices	Total (*n* = 366)	ADAM12 Expression		*p*-Value
		Negative (*n* = 201)	Positive (*n* = 165)	
KI (mean ± SD)	52.8 ± 15.1	51.1 ± 14.7	54.3 ± 15.6	0.504
AI (mean ± SD)	8.8 ± 5.6	11.3 ± 6.4	6.4 ± 3.7	0.009

ADAM12, A disintegrin and metalloprotease12; KI, Ki-67 labeling index; AI, apoptotic index; SD, standard deviation.

## Data Availability

Data available in a publicly accessible repository.

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
