# Peer review of "A Disintegrin and Metalloprotease 12 Promotes Tumor Progression by Inhibiting Apoptosis in Human Colorectal Cancer"

_cancers, 2021, doi:10.3390/cancers13081927_

Round 1

Reviewer 1 Report

None of the major points of critique in the previous review have been revised. 

Author Response

Thank you for your comments.

Reviewer 2 Report

Please, in the current format, the article could be published 

Best Regards,

Natalia Malara

Author Response

Thank you for your comments.

Reviewer 3 Report

The authors correctly addressed the requests made.

Author Response

Thank you for your comments.

The English of our manuscript was corrected by the English editing service provided by MDPI.

Reviewer 4 Report

Authors have addressed the issues raised. One comment:

The discussion would benefit from letting the reader know that you are aware of the differences between cell lines and colorectal cancer tissues

Previous comment: If not, please describe in the first section of the results why cell lines are selected lacking endogenous ADAM12 expression for the in-vitro experiments?

your comment: 

"We still don't know why ADAM12 expression is different from colorectal cancer tissue and colorectal cancer cell lines."

Author Response

Thank you for your comments. We edited the discussion section as followings : 

In our study, using human CRC tissues, although ADAM12 expression was not observed in CRC cell lines, ADAM12 expression was upregulated to higher levels in CRC tissues than in the normal mucosa counterparts at the mRNA and protein level in the colonoscopic biopsy and surgical specimens.

Also, The English of our manuscript was corrected by the English editing service provided by MDPI.

This manuscript is a resubmission of an earlier submission. The following is a list of the peer review reports and author responses from that submission.

Round 1

Reviewer 1 Report

The reported study investigates the ADAM12 protease in colon cancer. Using human colon cancer cells lines and a mouse xenograft model, the authors examined the effect of ADAM12 overexpression or knockdown (although this is not really a knockdown - see specific comments below) on cell proliferation, apoptosis and certain signal transduction pathways. Moreover, ADAM12 overexpression in tumor tissue from colon cancer patients was found to be positively correlated with disease stage and patient prognosis.

As several previous reports have demonstrated similar effects of ADAM12 in many different types of cancer, including colon cancer, the study provides little new knowledge. Importantly, as exemplified below, many conclusions are not supported by the data and the claimed aim to “elucidate whether a targeted downregulation of ADAM12 24 could lead to therapeutic sensitization” is not even assessed.

Specific comments:

- The authors claim that ADAM12 is upregulated in colorectal cancer, yet, they do not detect ADAM12 in the cell lines used. Thus, the experimental model system seems unjustified. Also, the authors knock down ADAM12 in cells transfected with ADAM12, but are lacking the condition where ADAM12 is knocked down without any prior overexpression.

- Western blots: no size-markers are shown; only one band of ADAM12 is seen (should be both a pro and mature form); no quantifications are provided (generally quantifications of at least 3 independent experiments should be shown).

- Experiments shown in Figure 2 and 3 do not support the conclusions. One examples of many: in Fig. 3, scrambled siRNA has an effect on its own. Also, how to explain that ADAM12 overexpression leads to increased cell proliferation AND cell cycle arrest?

- Fig. 4: Phosphorylated Akt and PTEN should be normalized to total Akt and PTEN levels.

- Important information is missing in the M&M section – e.g. mouse work, details on patients etc.  

- The authors describe the patient cohort as colorectal cancer (CRC) patients, but only colon cancer patients are included. Similarly, the cell lines are described as CRC cells – but are derived from colon cancer.  

Reviewer 2 Report

Journal Cancers

Manuscript ID: cancers-1043122 Title: Elucidation of the molecular mechanism of ADAM12 and its application as candidate biomarker in colorectal cancer.

In this manuscript the authors evaluated the roles of ADAM12 in colorectal cancer progression and prognosis by clarifying whether a targeted downregulation of ADAM12

could lead to therapeutic sensitization. While the text is interesting to read, there are some points that should be developed or improved before it can be considered publishable.

In particular below are listed the major revisions:

  1. the authors conduct their experimental flow using two secondary lines of colon cancer, SW480 and DLD1, i.e. epithelial cell lines obtained from patients with colon cancer in Duke B and C stage respectively. To achieve the purpose of the study it would be necessary to analyze further cell lines representative of the Duke A and D stages.

  1. flow cytometric analysis of the cell cycle phases distribution is the result of a statistical elaboration that the authors carried out using the Flow Jo software program, as reported in materials and methods, but they do not report correspondent CV for each analysis performed. CV or coefficient of variation, is an essential paramenter to understand if the distribution of the cell cycle phases has been consistent with the data collected and the standard.

  1. Furthermore, in the section on the effects of ADAM12 on apoptosis and cell cycle arrest in human CRC cells the apoptosis, are reported only the data that are obtained after transfection but there are no data relating to the apoptotic rate in the cells before each type of transfection. These data are necessary to individuate the apopoatotic rate, which is present in the culture under the same conditions (same passage and confluence) before each type of transfection performed. This result must be subtracted to those induced by specific transfection tested.

Minor revision:

some typos need to be fixed

Reviewer 3 Report

I read with interest the article by Park et al exploring the role of ADAM12 in CRC. Before a possible acceptance the authors should address the following minor points:

  • Check the English language.
  • Correct typos
  • Reset the layout of Table 1
  • Enrich the discussion section with comparisons of other findings of ADAM12 in other tumor settings.
  • Highlight the limits of the study
  • Write 2-3 bullet sentences on the future perspectives related to the real clinical application of exploring ADAM12 as biomarker of severity and invasiveness of CRC and as a terapheutic target

Reviewer 4 Report

Authors describe in-vitro experiments in which ADAM12 overexpression results in increased proliferation and decreased apoptosis of two cell lines. They show that ADAM12 is (over) expressed in human CRC and relate it to survival.

Introduction

Nice introduction, it could be strengthened by adding a few sentences adressing the networks in which ADAM12 is involved and its relationship to the genes described throughout the text (eg to akt and pten signaling)

Results

The first section needs  some clarity for the reader (at least for me..) I am a bit confused. SW480 is derived from a stage 3-4 adenocarcinoma and (as depicted in figure 1A) has no endogenic ADAM12 expression. Also, the DSD1 cel line has no endogenous ADAM12 protein expression. After transfection it does have ADAM10 expression and after sub sequent inhibition it does not. Line 76-78 ADAM12  protein expression increases upon transfection (as is depicted in the figure). Please rephrase line 81-83 to make this more clear (separate the sentences).  Please also elaborate on the fact why ADAM12 is not expressed in these cell lines. 

Interestingly, ADAM12 is expressed (and upregulated in tumors) in the human samples (fig5). Is ADAM10 expressed at low levels in the cell lines used? If not, please describe in the first section of the results why  cell lines are selected lacking endogenous ADAM12 expression for the in-vitro experiments?

Minor comments

figure 1 is small compared to the text and difficult to read. Authors might want to enlarge this figure.